# Evidence-based beta blocker use associated with lower heart failure readmission and mortality, but not all-cause readmission, among Medicare beneficiaries hospitalized for heart failure with reduced ejection fraction

**Matthew Shane Loop**[1]*, **Melissa K. Van Dyke**[2], **Ligong Chen**[3], **Todd M. Brown**[4], **Raegan W. Durant**[5], **Monika M. Safford**[6], **Emily B. Levitan**[3]

**1** Department of Biostatistics, University of North Carolina at Chapel Hill, Chapel Hill, North Carolina, United States of America, **2** Center for Observational Research, Amgen Inc., Thousand Oaks, California, United States of America, **3** Department of Epidemiology, University of Alabama at Birmingham, Birmingham, Alabama, United States of America, **4** Division of Cardiovascular Disease, University of Alabama at Birmingham, Birmingham, Alabama, United States of America, **5** Division of Preventive Medicine, University of Alabama at Birmingham, Birmingham, Alabama, United States of America, **6** Division of General Internal Medicine, Weill Cornell Medical College, Birmingham, Alabama, United States of America

* matthew_loop@unc.edu

## Abstract

The beta blockers carvedilol, bisoprolol, and sustained-release metoprolol succinate reduce readmissions and mortality among patients with heart failure with reduced ejection fraction (HFrEF), based upon clinical trial and registry studies. Results from these studies may not generalize to the typical patient with HFrEF. We conducted a retrospective cohort study of beneficiaries in the Medicare 5% sample hospitalized for HFrEF between 2007 and 2013 and were discharged alive. We compared the 30-day and 365-day heart failure (HF) readmission, all-cause readmission, and mortality rates between beneficiaries who filled a prescription for an evidence-based beta blocker and those who did not after being hospitalized for HFrEF. Out of 12,127 beneficiaries hospitalized for HFrEF, 20% were readmitted for HF, 62% were readmitted for any cause, and 27% died within 365 days. In competing risk models adjusted for demographics, healthcare utilization, and comorbidities, beta blocker use was associated with a lower risk of HF readmission between 8–365 days post discharge (hazard ratio 0.79 [95% confidence interval 0.76, 0.82]), but was not significantly associated with all-cause readmission (1.02 [0.97–1.07]). In Cox models adjusted for the same covariates, beta blocker use was associated with lower mortality 8–365 days post discharge (0.65 [0.60–0.71]). Results were similar when follow up was truncated at 30 days post discharge. Increasing the use of beta blockers following HFrEF hospitalization may not decrease all-cause readmissions among Medicare beneficiaries, but may reduce HF-specific readmissions and mortality.

**Data Availability Statement:** The data contain confidential information concerning Medicare beneficiaries. Therefore, we do not have the authority to share them per the restrictions imposed by the Centers for Medicare and Medicaid Services. Requests may be sent to ResDAC (https://www.resdac.org/; resdac@umn.edu) in order to obtain the Medicare data.

**Funding:** The study was funded by Amgen Inc and NIH National Heart, Lung, and Blood Institute (5T32HL00745734). This research was supported by an academic collaboration between the University of Alabama at Birmingham and Amgen, Inc. The funders provided comments on the design and interpretation of this work. The academic authors conducted all analyses and maintained the rights to publish this manuscript.

**Competing interests:** At the time the research was primarily conducted, Dr. Loop received salary support from Amgen Inc. Dr. Van Dyke was employed in the Center for Observational Research, Amgen Inc. during the time the research was conducted. Dr. Chen, Dr. Brown, Dr. Durant, and Dr. Levitan have received research grants from Amgen Inc. Dr. Levitan serves on the Advisory Board for Amgen Inc. This does not alter our adherence to PLOS ONE policies on sharing data and materials.

# Introduction

The beta blockers carvedilol, bisoprolol, and sustained-release metoprolol succinate have been shown to reduce readmissions and mortality for patients with heart failure with reduced ejection fraction (HFrEF). These findings have led to the use of beta blocker prescriptions at hospital discharge as an indicator of quality of care [1,2]. However, the evidence for their benefit is based largely upon patients enrolled in randomized controlled trials (RCTs) and heart failure registries [3–6]. Participants in these studies were likely different from the typical patient hospitalized for HFrEF, potentially limiting the generalizability of these results. For example, only 25% of Medicare beneficiaries hospitalized with HFrEF would have met the inclusion criteria for the RCTs due to age, contraindications, or comorbidities [7]. Additional therapies such as eplerenone [8] and sacubitril/valsartan [9] have demonstrated effective reduction in mortality and readmissions, but sacubitril/valsartan had yet to experience meaningful uptake in its first 18 months of approval through the end of 2016 [10]. Beta blocker therapy, in addition to other therapies such as ACE-inhibitors, remain common pharmaceutical therapies for HFrEF in the general patient population. Therefore, we conducted a retrospective cohort study of the Medicare 5% random sample to determine the relative risk for heart failure (HF) readmission, all-cause readmission, and mortality associated with use of these evidence-based beta blockers among beneficiaries hospitalized for HFrEF.

# Materials and methods

## Study sample

Using the 5% random sample of Medicare beneficiaries, we identified beneficiaries who: had an inpatient claim and were discharged alive with a primary discharge diagnosis of 428.2x (systolic heart failure) or 428.4x (combined systolic and diastolic heart failure) between 2007 and 2013, were living in the US for 365 days prior to hospital admission date, had continuous Medicare Part A, B, and D coverage for 365 days prior to hospital admission date, had continuous inclusion in the Medicare 5% sample for 365 days prior to hospital admission date, and were less than 110 years on hospital admission date. Follow up data after hospital discharge were collected through one year, with follow up data available through 2013. We conducted analyses with 30 days of follow up and with 365 days of follow up. We used the first eligible HFrEF hospitalization for each beneficiary as the index hospitalization, which was not necessarily an incident HFrEF hospitalization, and we excluded beneficiaries who were discharged to a skilled nursing facility (SNF) because medications are not generally billed separately during a SNF stay [11].

## Beta blocker prescription fills

Previous studies with registry populations have relied upon discharge prescriptions in the medical record [3]. However, discharge prescriptions may be an indicator of provider behavior rather than patient behavior. Therefore, we used a claim for filling a prescription for an evidence-based beta blocker for HFrEF as a proxy for taking an evidence-based beta blocker at least once within a year following hospitalization. Prescription fills for carvedilol, bisoprolol, or sustained-release metoprolol succinate were identified using Medicare Part D pharmacy claims. Beneficiaries were considered non-users until they had a claim for a prescription fill. Once a person had a claim for a prescription fill, they were considered a user for the remainder of follow up.

## Outcomes

The primary outcomes of interest in this study were: (1) time to HF readmission (defined as a hospitalization claim with a primary discharge diagnosis of HF, with or without indication of systolic function; see S1 Methods); (2) time to all-cause readmission; and (2) time to death. We

used follow up at 30 days and 365 days as the time intervals of interest. Beneficiaries were censored if they lost fee-for-service coverage in Medicare, left the 5% national sample, or moved out of the 50 United States or Washington D.C. Death was considered to be a competing risk for readmission. All-cause mortality was collected from the Medicare beneficiary enrollment file that includes death dates reported by the Social Security Administration.

## Covariates

To account for potential confounding, we included demographics, healthcare utilization, comorbidities, use of other medications for HF, and year of HFrEF hospitalization in our multivariable-adjusted models. Demographic variables included age at hospital admission, sex, race (black, other, and white), US census region of residence (East North Central, East South Central, Middle Atlantic, Mountain, New England, Pacific, South Atlantic, West North Central, and West South Central), dual-eligibility for Medicare and Medicaid, and having a Medicare Part D subsidy. Healthcare utilization during the year prior to hospitalization was captured by whether the beneficiary had a prescription fill for a beta blocker (evidence-based beta blocker for HFrEF, other beta blocker, or none), was hospitalized for any cause, lived in a nursing home [12], or had a stay in a SNF. Comorbidities during the year prior to hospitalization included anemia, atrial fibrillation, COPD, the Charlson comorbidity index, depression, hypotension, liver disease, and malnutrition (see S1 Methods for specific definitions). Use of other medications for HF in the past year include ACEI/ARB use and diuretic use.

## Statistical analysis

First, we calculated summary statistics for each covariate for those that ever filled a prescription for an evidence-based beta blocker within 30 days and for those that filled a prescription for an evidence-based beta blocker after 30 days or never filled a prescription. Second, we fit a Fine and Gray [13] competing risk model for the readmission outcomes (HF and all-cause), in order to account for the important competing risks of mortality and other types of readmission (when HF readmission was the outcome). We fit a standard Cox proportional hazards model for mortality. Competing risk models allowed for comparisons of readmission outcomes after accounting for potential differences in follow up time due to a competing outcome between those who used beta blockers versus those who did not. Beta blocker use was modeled as a time-varying exposure. Beneficiaries were considered to be beta blocker users from the date of the first fill through the end of follow up. We allowed the risk ratio (RR) for beta blocker use to vary with time by using an interaction with a categorical variable for time post hospital discharge (0–3 days, 4–7 days, or >= 8 days), in order to satisfy the proportional hazards assumption [14]. We reported the RRs for each of these time intervals.

We conducted four sensitivity analyses. First, we repeated the analysis for a subset of beneficiaries that had characteristics similar to an analysis from the Organized Program to Initiate Lifesaving Treatment in Hospitalized Patients with Heart Failure (OPTIMIZE-HF) registry [3], which was limited to patients with HFrEF who were naive to beta blocker treatment (defined as not filling a prescription for any beta blocker in the year prior to HFrEF hospitalization), were at least 65 years of age, and were good candidates for beta blockers by excluding beneficiaries with either bradycardia or atrioventricular block (degree two or three) but without an accompanying pacemaker, had asthma, or had hypotension, all in the year prior to hospitalization for HFrEF, were discharged to hospice or against medical advice, had cardiogenic shock during the hospitalization for HFrEF, or were transferred to another acute care facility. Second, we conducted an analysis that excluded beneficiaries with any days of carvedilol, bisoprolol, or sustained-release metoprolol succinate available on the day of hospital admission

according to Part D pharmacy claims for the beneficiary, in order to decrease the potential for exposure misclassification for participants who might have been taking beta blockers from a previous prescription immediately following discharge. Third, we repeated the analysis in a subset of participants who did not have history of a procedure code for an implanted cardiac device (pacemaker and/or defibrillator; see S1 Methods for codes used). Potential contraindications or variation in effects of beta blockers between those with and without an implanted cardiac device motivated this analysis. Fourth, we modified the exposure from "filled or not filled" to "filled carvedilol, filled bisoprolol, filled sustained-release metoprolol succinate, or not filled." The first fill for one of these beta blockers determined the exposure category in which the beneficiary remained during the rest of follow up, regardless of future beta blocker fills. This sensitivity analysis attempted to detect any variation in the associations between fills for specific beta blockers and the outcomes of interest. For all sensitivity analyses, we reported only the hazard ratios for the 8–30 or 8–365 day time period.

This study was approved by the University of Alabama at Birmingham Institutional Review Board and the Centers for Medicare and Medicaid Services Privacy Board.

## Results

These inclusion / exclusion criteria led to a final sample size of 12,127 beneficiaries hospitalized for HFrEF. Although our exclusion criteria were necessary to identify our target study population, they removed 80% of the identified HF hospitalizations from 2007 to 2013 among beneficiaries in our 5% random sample (see S1 Table). Many beneficiaries were excluded because the HF diagnosis code did not contain information on systolic function or indicated isolated diastolic dysfunction. The median(interquartile range [IQR]) days of follow up for the 365-day interval was 136(332) for HF readmission, 98(261) days for all-cause readmission, and 365(210) for mortality. By 365 days, 62% of beneficiaries were readmitted and 27% died. Forty-three percent of beneficiaries filled a prescription for an evidence-based beta blocker within 30 days of discharge. The median(IQR) time to prescription fill among those who filled a prescription for an evidence-based beta blocker within 30 days was 5(24) days. Summaries of demographics, healthcare utilization, comorbidities, potential contraindications, and year of hospitalization by whether the beneficiary filled a prescription for an evidence-based beta blocker within 30 days are shown in Table 1.

Multivariable-adjusted RRs for readmission and mortality are shown in Table 2/Fig 1. In fully adjusted models, filling a prescription for an evidence-based beta blocker was associated with an approximately 20% lower risk of an HF readmission from 8–30 and 8–365 days post discharge. However, filling a prescription for an evidence-based beta blocker was not significantly associated with all-cause readmission from 8–30 or 8–365 days, with RRs close to the null value of 1. Filling a prescription for a beta blocker was associated with a 32% lower mortality from 8–30 days and a 35% lower mortality from 8–365 days post discharge.

S2 Table shows the sample characteristics for the sensitivity analysis of the subsample similar to OPTIMIZE-HF that excluded 10,514 beneficiaries, and S3 Table shows the multivariable-adjusted RRs for this subsample. In the subsample similar to OPTIMIZE-HF, beta blocker use was associated with a 5% lower risk of HF readmission from 8–30 days, but was not significantly associated with HF readmission from 8–365 days, with a RR of 1. Similar to the main analysis, the association between beta blocker use and all-cause readmission was not statistically significant from 8–30 or 8–365 days. The association with mortality was not statistically significant from 8–30 days with a wide confidence interval, likely due to a smaller number of events. However, beta blocker use was associated with a statistically significant 36% lower mortality at 8–365 days post discharge in the sample similar to OPTIMIZE-HF.

**Table 1. Characteristics of medicare beneficiaries hospitalized for HFrEF by whether beneficiary filled a prescription for an evidence-based beta blocker within 30 days.**

| Variable | Level | Filled prescription for beta blocker within 30 days (n = 5,164) | Did not fill prescription for beta blocker within 30 days (n = 6,963) |
|---|---|---|---|
| Age at admission (years)[a] | | 74.1 (12.1) | 75.8 (12.3) |
| Race | Black | 889 (17.2) | 1002 (14.4) |
| | Other | 322 (6.2) | 397 (5.7) |
| | White | 3953 (76.5) | 5564 (79.9) |
| Women | | 2533 (49.1) | 3437 (49.4) |
| Dual-eligible for Medicare and Medicaid | | 2112 (40.9) | 2547 (36.6) |
| Medicare Part D subsidy | | 2558 (49.5) | 3114 (44.7) |
| US Census region | East North Central | 880 (17.0) | 1215 (17.4) |
| | East South Central | 534 (10.3) | 731 (10.5) |
| | Middle Atlantic | 644 (12.5) | 1054 (15.1) |
| | Mountain | 195 (3.8) | 281 (4.0) |
| | New England | 233 (4.5) | 334 (4.8) |
| | Pacific | 533 (10.3) | 589 (8.5) |
| | South Atlantic | 1002 (19.4) | 1344 (19.3) |
| | West North Central | 454 (8.8) | 487 (7.0) |
| | West South Central | 689 (13.3) | 928 (13.3) |
| Anemia | | 2598 (50.3) | 3929 (56.4) |
| Asthma | | 801 (15.5) | 1103 (15.8) |
| Atrial fibrillation | | 2132 (41.3) | 3415 (49.0) |
| Atrioventricular block (2nd or 3rd degree) | | 36 (0.7) | 41 (0.6) |
| Beta blocker use at baseline | Evidence-based | 3305 (64.0) | 2907 (41.7) |
| | None | 1165 (22.6) | 1851 (26.6) |
| | Other beta blocker | 694 (13.4) | 2205 (31.7) |
| ACEI/ARB use | | 3690 (71.5) | 4666 (67.0) |
| Diuretic use | | 3879 (75.1) | 5264 (75.6) |
| Bradycardia | | 55 (1.1) | 127 (1.8) |
| COPD | | 2334 (45.2) | 3425 (49.2) |
| Cardiogenic shock | | 41 (0.8) | 57 (0.8) |
| Charlson comorbidity index | 0 | 1914 (37.1) | 2704 (38.8) |
| | 1–3 | 723 (14.0) | 827 (11.9) |
| | >=4 | 2527 (48.9) | 3432 (49.3) |
| Depression | | 1012 (19.6) | 1431 (20.6) |
| Discharged against medical advice | | 26 (0.5) | 42 (0.6) |
| Discharged to hospice | | 50 (1.0) | 449 (6.4) |
| Hospitalization during baseline | | 1273 (24.7) | 2088 (30.0) |
| Hypotension | | 968 (18.7) | 1465 (21.0) |
| Liver disease | | 225 (4.4) | 344 (4.9) |
| Malnutrition | | 277 (5.4) | 545 (7.8) |
| Nursing home residence | | 283 (5.5) | 588 (8.4) |
| Skilled nursing facility stay | | 473 (9.2) | 959 (13.8) |

*(Continued)*

**Table 1.** (Continued)

| Variable | Level | Filled prescription for beta blocker within 30 days (n = 5,164) | Did not fill prescription for beta blocker within 30 days (n = 6,963) |
|---|---|---|---|
| Year of hospitalization | 2007 | 279 (5.4) | 336 (4.8) |
| | 2008 | 741 (14.3) | 901 (12.9) |
| | 2009 | 776 (15.0) | 1047 (15.0) |
| | 2010 | 853 (16.5) | 1115 (16.0) |
| | 2011 | 827 (16.0) | 1161 (16.7) |
| | 2012 | 815 (15.8) | 1172 (16.8) |
| | 2013 | 873 (16.9) | 1231 (17.7) |

[a]mean (standard deviation)

The sensitivity analysis that excluded participants with any days of an evidence-based beta blocker available upon hospital admission (S4 Table) produced estimates of the association between beta blocker use and the outcomes that were similar to the main analysis. The sensitivity analysis that excluded participants with an implanted cardiac device produced estimates almost identical to the main analysis as well (S5 Table). The final sensitivity analysis, which specified which particular evidence-based beta blocker was filled (carvedilol, bisoprolol, metoprolol succinate, or none), showed nearly identical RRs for each specific beta blocker vs. no fill (S1 Fig). The confidence intervals for the RRs for bisoprolol fill vs. no fill were sometimes much larger than for the other beta blockers, indicating fewer beneficiaries filling prescriptions for bisoprolol compared to the carvedilol or sustained-release metoprolol succinate.

## Discussion

We identified a cohort of Medicare beneficiaries hospitalized with a primary discharge diagnosis of HFrEF, with the goal of comparing readmission rates among those who filled a

**Table 2.** Risk ratios (RRs) and 95% confidence intervals for filling a prescription for carvedilol, bisoprolol, or sustained-release metoprolol succinate after discharge from a hospitalization for heart failure with reduced ejection fraction (HFrEF).

| Outcome[a] | Time period within follow up | 30 days | 365 days |
|---|---|---|---|
| HF readmission | 0–3 days | 0.92 (0.91–0.93) | 0.89 (0.87–0.90) |
| HF readmission | 4–7 days | 0.97 (0.96–0.98) | 0.95 (0.93–0.96) |
| HF readmission | >7 days | 0.83 (0.81–0.84) | 0.79 (0.76–0.82) |
| All-cause readmission | 0–3 days | 1.08 (0.84–1.39) | 1.08 (0.84–1.39) |
| All-cause readmission | 4–7 days | 1.08 (0.88–1.32) | 1.05 (0.86–1.28) |
| All-cause readmission | >7 days | 0.97 (0.88–1.08) | 1.02 (0.97–1.07) |
| Mortality | 0–3 days | 0.17 (0.08–0.37) | 0.17 (0.08–0.37) |
| Mortality | 4–7 days | 0.38 (0.22–0.68) | 0.37 (0.21–0.66) |
| Mortality | >7 days | 0.69 (0.55–0.86) | 0.65 (0.60–0.71) |

[a]Models were adjusted for age at admission, sex, race, US census region, year of HFrEF hospitalization, as well as several variables assessed during the year prior to hospitalization: type of beta blocker use (evidence-based beta blocker for HFrEF, any other beta blocker, or none), ACEI/ARB use, diuretic use, dual-eligibility, Medicare Part D subsidy, nursing home residence, atrial fibrillation, malnutrition, liver disease, anemia, depression, COPD, Charlson comorbidity index, hospitalization, and a skilled nursing facility (SNF) stay. An RR of 1 indicated no significant association

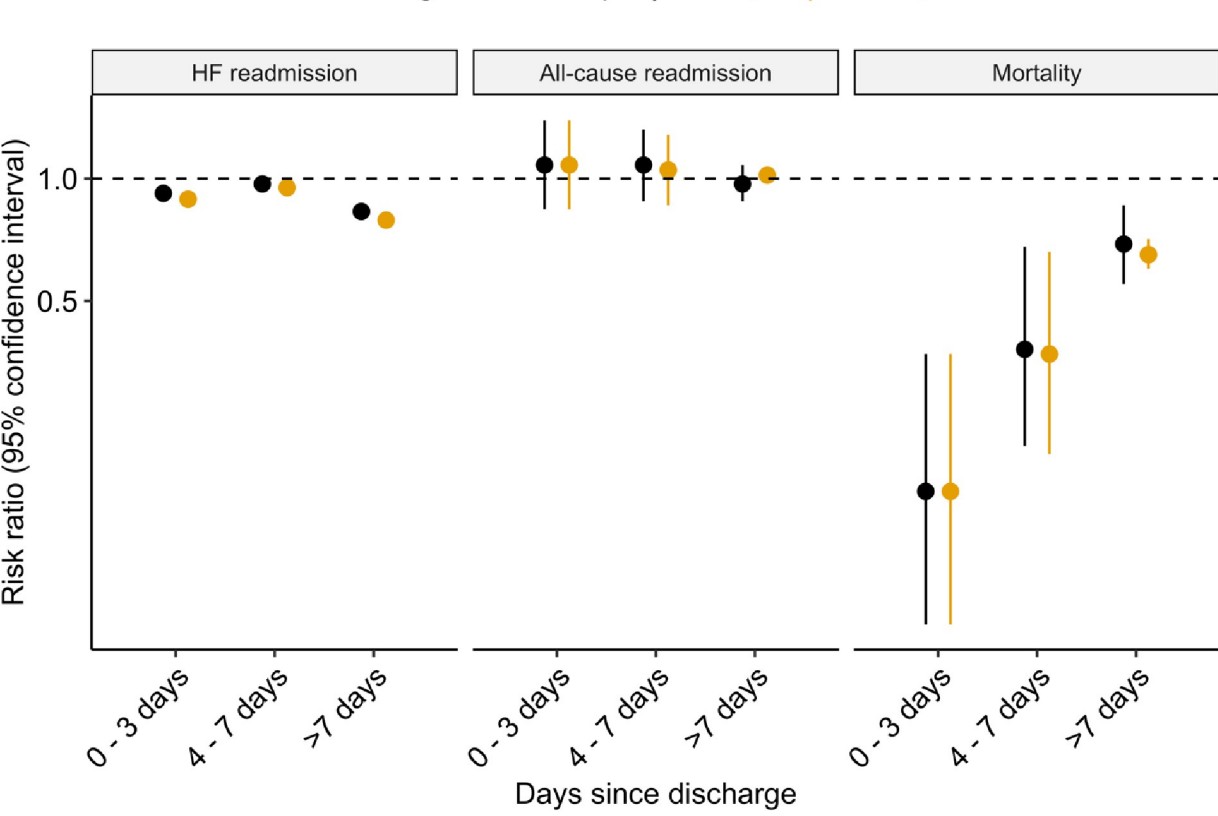

**Fig 1. Risk ratios (RRs) and 95% confidence intervals for filling a prescription for carvedilol, bisoprolol, or sustained-release metoprolol succinate after discharge from a hospitalization for heart failure with reduced ejection fraction (HFrEF).** Models were adjusted for age at admission, sex, race, US census region, year of HFrEF hospitalization, as well as several variables assessed during the year prior to hospitalization: type of beta blocker use (evidence-based beta blocker for HFrEF, any other beta blocker, or none), ACEI/ARB use, diuretic use, dual-eligibility, Medicare Part D subsidy, nursing home residence, atrial fibrillation, malnutrition, liver disease, anemia, depression, COPD, Charlson comorbidity index, hospitalization, and a skilled nursing facility (SNF) stay. An RR of 1 indicated no significant association. Although confidence intervals are plotted, the intervals are so narrow that some are hard to see.

prescription for an evidence-based beta blocker for HFrEF and those who did not. We used prescription fill claims as a proxy for beta blocker use. We found that beta blocker use was associated with lower risk of HF readmission at 30 and 365 days, after accounting for the competing risks of readmissions for other causes and mortality. We found that the 30-day and 365-day all-cause readmission rates were similar between beneficiaries who filled a prescription for an evidence-based beta blocker and those who did not, after accounting for the increased mortality among those that did not fill a prescription. Finally, mortality was approximately 30%–35% lower among those that filled a prescription for an evidence-based beta blocker.

In contrast to OPTIMIZE-HF, our main analysis did not find evidence of an association between beta blocker use and all-cause readmission, after accounting for the increased mortality among those that did not use a beta blocker. One explanation for the differences could be that Hernandez et al. (2009) [3] assessed use of any beta blocker, as opposed to evidence-based beta blockers for HFrEF. We observed similar 1-year readmission rates (64% in OPTIMIZE-HF vs. 62% in our study) and mortality rates (33% in OPTIMIZE-HF vs. 27% in our study). When we restricted our study sample to beneficiaries that met exclusion criteria similar to OPTIMIZE-HF, the estimated RRs were similar to those in the main analysis for

readmission and mortality at both 30 days and 365 days post hospital discharge. When we considered only HF readmissions in the main sample, we found RRs similar to the RRs for all-cause readmission in OPTIMIZE-HF (0.89 for 1-year all-cause readmission in OPTIMIZE-HF vs. 0.83 for 30-day HF readmission and 0.79 for HF readmission at 365 days in our study). A study of Medicare beneficiaries in Alabama found an association between beta blocker prescriptions and mortality, but not HF readmission or all-cause readmission [15]. However, this study was small (380 matched pairs) and did not take into account competing risks.

Among Medicare beneficiaries with chronic heart failure, 45% of preventable hospital admissions over a 1-year period are due to non-cardiac causes, and the presence of non-cardiac comorbidities is associated with increased risk of readmission and mortality [16]. Medicare beneficiaries might be readmitted for non-HF related conditions so often that beta blockers do not have a significant benefit on readmissions, even though beta blockers improve heart function [17]. In both OPTIMIZE-HF [18] and our study population [19], approximately half of readmissions before 30 days were cardiovascular-related. We found that 20% of readmissions were non-HF cardiovascular readmission. Although the three beta blockers we studied are preferred in HFrEF because of their HFrEF-specific benefits, the other beta blockers may have beneficial effects on other non-HF cardiovascular causes of hospitalization. This possibility could help to explain the differences in finding between our study and OPTIMIZE-HF. Increasing the number of filled beta blocker prescriptions among patients hospitalized for HFrEF might not reduce total readmission rates among typical patients with HFrEF. However, we did still observe a benefit for mortality.

## Limitations and strengths

Although our study addressed gaps in our understanding of the effects of beta blockers in typical patients with HFrEF, it had limitations. First, because of the time period of our study, we were not able to assess the impact of newly recommended angiotensin receptor-neprilysin inhibitors (ARNI) or sinoatrial node modulators (i.e., ivabradine) in Medicare beneficiaries [20,21]. Additional research with sufficient follow up time of Medicare beneficiaries using these medications will be critical for understanding these medications' population-based impact. We did not include measures of dosage, but previous studies have found that few Medicare beneficiaries with HFrEF are uptitrated on an evidence-based beta blocker within one year of discharge [22]. We assumed that filling a prescription for a beta blocker was a proxy for continued use for the remainder of the period of follow up, but some participants will have stopped taking the medication during follow up in response to potential side effects or new contraindications. Therefore, out estimates of benefit may be exaggerated or underestimated compared to the true benefit. Perhaps the most consequential limitation of our study was that we could not fully account for the propensity of the sickest patients with HFrEF to not receive treatment (i.e., "confounding by indication") [23], due to lack of access to vital signs, laboratory measures, cardiac function (e.g., ejection fraction), and patient preference in Medicare data to calculate and adjust for an appropriate risk score [24]. Another source of confounding could have been that patients who are more likely to be adherent to any drug regimen for HFrEF and thus have better outcomes may be more likely to fill a prescription for a beta blocker. Therefore, this analysis may overestimate the apparent benefits of beta blocker use. In particular, filling a prescription for a beta blocker shortly after discharge may be a marker of adherence to medical therapy in general, which may help explain the large apparent benefit on mortality immediately following hospitalization. Finally, even though our study targeted a more general patient population than previous RCTs of beta blockers or registry studies, we were still somewhat limited in our ability to generalize our results to all Medicare

beneficiaries. Participants who were excluded from the study sample were more likely to be female, live in a nursing home, and have had a SNF stay in the year prior to hospitalization and less likely to be taking an evidence-based beta blocker, which were all expected differences given our exclusion criteria (see S1 Table). However, beneficiaries excluded from our analysis were also more likely to be dual-eligible for Medicare and Medicaid, have a Medicare Part D subsidy, and have a higher comorbidity burden. We therefore implicitly excluded sicker beneficiaries of lower socioeconomic status, and it is possible that these beneficiaries would have been less likely to fill a prescription for a beta blocker and have higher risk of the outcomes of interest. It is unclear whether our estimates generalize to these beneficiaries.

Strengths of our study included the use of a large and more generalizable sample of patients with HFrEF than previous studies, using claims for prescription fills, which are more proximal to medication used compared to discharge prescriptions, and using a claims-based definition of HFrEF that has a positive predictive value of 77% [25] m as well as using statistical methods to reduce bias including accounting for competing risks in our survival analysis models, accounting for non-proportional hazards, and using prescription fills for a beta blocker after hospitalization as a time-varying covariate.

## Conclusions

In conclusion, among Medicare beneficiaries with HFrEF, we found no evidence that evidence-based beta blocker use associated with all-cause readmission after adjusting for differences in mortality. Beta blocker use was associated with lower mortality and lower risk of HF readmission. However, in this retrospective observational study, we were unable to disentangle the effects of beta blocker use from other characteristics and behaviors such as adherence to other therapies. This adherence to other therapies may have led to overestimation of the protective associations of beta blocker use, particularly on mortality. Increasing the use of beta blockers following HFrEF hospitalization is unlikely to decrease all-cause readmissions among Medicare beneficiaries, but may reduce HF-specific readmissions and mortality.

## Supporting information

**S1 Method.**
(DOCX)

**S1 Fig. Risk ratios (RRs) and 95% confidence intervals for filling a prescription for a specific beta blocker (carvedilol, bisoprolol, or sustained-release metoprolol succinate) vs. no fill after discharge from a hospitalization for heart failure with reduced ejection fraction (HFrEF).**
(DOCX)

**S1 Table. Summary of beneficiaries with a hospitalization for HFrEF between 2007 and 2013 who were and were not included in the current analysis.**
(DOCX)

**S2 Table. Summary of subsample similar to OPTIMIZE-HF by whether beneficiary filled a prescription for an evidence-based beta blocker within 30 days.**
(DOCX)

**S3 Table. Risk ratios (RRs) and 95% confidence intervals for filling a prescription for an evidence-based beta blocker (carvedilol, bisoprolol, or sustained-release metoprolol succinate) after discharge from a hospitalization for heart failure with reduced ejection fraction**

(HFrEF), among those in a subsample similar to the OPTIMIZE-HF cohort[a].
(DOCX)

**S4 Table. Risk ratios (RRs) for readmission and all-cause mortality comparing those filling a prescription for an evidence-based beta blocker (carvedilol, bisoprolol, or sustained-release metoprolol succinate) after discharge from a hospitalization for heart failure with reduced ejection fraction (HFrEF), among those with no evidence-based beta blockers available upon hospital admission.**
(DOCX)

**S5 Table. Risk ratios (RRs) for readmission and all-cause mortality comparing those filling a prescription for an evidence-based beta blocker (carvedilol, bisoprolol, or sustained-release metoprolol succinate) after discharge from a hospitalization for heart failure with reduced ejection fraction (HFrEF), among those with no implanted cardiac device.**
(DOCX)

## Author Contributions

**Conceptualization:** Melissa K. Van Dyke, Ligong Chen, Todd M. Brown, Raegan W. Durant, Monika M. Safford, Emily B. Levitan.

**Data curation:** Ligong Chen, Emily B. Levitan.

**Formal analysis:** Matthew Shane Loop.

**Funding acquisition:** Emily B. Levitan.

**Investigation:** Matthew Shane Loop.

**Methodology:** Matthew Shane Loop, Emily B. Levitan.

**Supervision:** Emily B. Levitan.

**Visualization:** Matthew Shane Loop.

**Writing – original draft:** Matthew Shane Loop, Ligong Chen.

**Writing – review & editing:** Matthew Shane Loop, Melissa K. Van Dyke, Todd M. Brown, Raegan W. Durant, Monika M. Safford, Emily B. Levitan.

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
