## [Decision Letter · Decision Letter 0]

26 Sep 2019

PONE-D-19-24254

Evidence-based beta blocker use associated with lower heart failure readmission and mortality, but not all-cause readmission, among Medicare beneficiaries hospitalized for heart failure with reduced ejection fraction

PLOS ONE

Dear Dr. Loop,

Thank you for submitting your manuscript to PLOS ONE. After careful consideration, we feel that it has merit but does not fully meet PLOS ONE’s publication criteria as it currently stands. Therefore, we invite you to submit a revised version of the manuscript that addresses the points raised during the review process.

Important points to address are the consideration of adjusting the analysis for other drugs, the discussion based on current treatment guidelines and the appropriate discussion of the limitations. In addition, I am wondering if the authors provide information as to whether there was a shift from less HF hospitalizations to an increase in non-HF hospitalizations and if yes, which causes of hospitalization increased, or if the lack of effect on all-cause hospitalization was simply because non-HF / non-cardiac hospitalizations were much more common that the effect on HF-hospitalization was diluted.

We would appreciate receiving your revised manuscript by Nov 10 2019 11:59PM. To enhance the reproducibility of your results, we recommend that if applicable you deposit your laboratory protocols in protocols.io, where a protocol can be assigned its own identifier (DOI) such that it can be cited independently in the future. For instructions see: http://journals.plos.org/plosone/s/submission-guidelines#loc-laboratory-protocols

We look forward to receiving your revised manuscript.

Kind regards,

Hans-Peter Brunner-La Rocca, M.D.

Academic Editor

PLOS ONE

Journal Requirements:

2. In ethics statement in the manuscript and in the online submission form, please provide additional information about the patient records used in your retrospective study. Specifically, please ensure that you have discussed whether all data were fully anonymized before you accessed them and/or whether the IRB or ethics committee waived the requirement for informed consent. If patients provided informed written consent to have data from their medical records used in research, please include this information.

Dr. Loop has received salary support from Amgen Inc. Dr. Van Dyke previously worked in the Center for Observational Research, Amgen Inc. Dr. Chen, Dr. Brown, Dr. Durant, and Dr. Levitan have received research grants from Amgen Inc. Dr. Levitan serves on the Advisory Board for Amgen Inc.  

We note that one or more of the authors are employed by a commercial company: Amgen Inc

Additional Editor Comments (if provided):

Reviewers' comments:

Reviewer's Responses to Questions

**Comments to the Author**

1. Is the manuscript technically sound, and do the data support the conclusions?

Reviewer #1: Yes

Reviewer #2: Partly

2. Has the statistical analysis been performed appropriately and rigorously? 

Reviewer #1: Yes

Reviewer #2: Yes

3. Have the authors made all data underlying the findings in their manuscript fully available?

Reviewer #1: Yes

Reviewer #2: Yes

4. Is the manuscript presented in an intelligible fashion and written in standard English?

Reviewer #1: Yes

Reviewer #2: Yes

5. Review Comments to the Author

Reviewer #1: General comment, summary

The manuscript entitled “Evidence-based beta blocker use associated with lower heart failure readmission and mortality, but not all-cause readmission, among Medicare beneficiaries hospitalized for heart failure with reduced ejection fraction” by Loop et al. reported from a

retrospective cohort study of beneficiaries in the Medicare 5% sample hospitalized for

HFrEF between 2007 and 2013 and were discharged alive. They compared in total 12,127 participants, the 30-day and 365-day heart failure (HF) readmission, all-cause readmission, and mortality rates between beneficiaries who filled a prescription for an evidence-based beta blocker (carvedilol, bisoprolol, and sustained-release metoprolol succinate) and those who did not after being hospitalized for HFrEF.

The main finding of the analyses (adjusted risk models) and conclusion accordingly, was that increasing the use of beta blockers following HFrEF hospitalization is unlikely to decrease all-cause readmissions among Medicare beneficiaries, but may reduce HF-specific readmissions and mortality.

In my perspective, this analysis from a large HFrEF cohort, though retrospective, is well-conducted and the results are clinically important and add to our understanding of relevant outcomes associated with evidence-based HF therapies in real-world practice, e.g. in US.

The authors also included two sensitivity analyses in their report.

The present manuscript is easy-to-follow, well written, both in wording and grammar.

The comprehensive tables and figure, respectively contribute to the clear presentation of the results and interpretation.

Major comments

1. This retrospective cohort study is observational with its inherent limitations, although is a sample of real-world HF care in the US. The cohort, however, is not that contemporary.

2. Definition of HFrEF is not clear, uniform, or at discretion of the physician. Although, the sample is random and for that reason, close to generalizability.

3. Information on optimal or maximal beta-blocker doses is not provided (or lacking, not collected?); suboptimal up-titrated medication may be associated with more severe HFrEF, comorbidity burden, high age, and many other factors; and has effects on important morbidity and mortality.

4. Confounding by indication could only be partially accounted for.

5. Do the authors have insights or due thoughts on (substantially reported) non-presciption of beta-blockers?

6. Differential analyses of each beta-blocker may add to our knowledge, e.g. carvedilol as additional pharmocodynamic properties.

7. Do the authors have data on repeat hospitalizations?

8. Do the authors have data on (CIED) device usage; which may have effects on clinical outcomes? This also goes for non-HF cardiovascular medication.

9. How do the authors explain their finding that all-cause hospitalization is not significantly associated with filled prescription of a disease-modifying beta-blocker? In contrast to OPTIMIZE-HF, although in which all beta-blockers were taken into analysis?

10. The latter finding is strikingly important, is even incorporated in the manuscript title, and needs more detailed clarification and discussion accordingly, and guidance for future studies.

Minor comments

1. What definitions of anemia, atrial fibrillation and other comorbidities were used?

2. The authors used claim for a prescription fill as a proxy for beta-blocker use by HFrEF patients; has that been validated previously?

3. There is no certainty at all for continued use of beta-blocker, discontinuation, or intolerance, and also, no reported reasons for modified medication regimens

4. Relatively more patients from lower socio-economic status were excluded, which may influence outcomes and also elaborated risk stratification and generalizability.

5. The authors may describe strengths and limitations in a more separate paragraph or section at the end of Discussion, just prior to Conclusions.

6. In Figure 1, first panel, CIs are not depicted.

Reviewer #2: Summary:

This retrospective study investigated the effect of beta-blocker treatment in HFrEF beneficiaries in the Medicare 5% sample. For study inclusion, HFrEF patients had to have HFrEF related HF hospitalized between 2007 and 2013 and had to be alive at discharge.

Beta-blocker treatment after hospital discharge was assumed on the basis that study participants (overall 35%) filled prescription of an evidenced-based beta blocker (metoprolol, bisoprolol, carvedilol). These study participants were compared to study participants who did not fill a prescription for beta blocker treatment after hospital discharge. Outcome measure was 30 day and 365 day heart failure or all cause readmission rate, or all cause mortality. The first occurring event was always entered into the analysis. Out of 12127 beneficiaries with index hospitalization for HFrEF 30% were readmitted for HF hospitalization, 62% for any cause; 27% died within 365 days.

Filling a prescription for a beta blocker was associated with a lower HF readmission rate and a lower mortality rate but did not change the all-cause readmission rate in competing risk models.

Critique:

This retrospective analysis of data derived from U.S. medicare beneficiaries shows beneficial effects of beta-blocker treatment on heart failure readmission and mortality when study participants filled the beta blocker prescription within 0-3 day, 4-7 days, or > 7 days. This beneficial effect was already present within the first 30 days and persisted thereafter. This observation is surprising since Kaplan-Meier curves presenting survival or the combined endpoint of survival and heart-failure related hospitalization separated only after 3 months beta blocker treatment in the MERIT-HF or the COPERNICUS trial. Likewise, separation of survival curves in the BIOSTAT-CHF study occurred only after 100 days in patients on guidelines-based medical therapy. I wonder whether this important early effect is really associated with beta blocker treatment alone and not related to overall good compliance to drug treatment in the 35% of study participants filling the prescription?

The authors should discuss why the results of this retrospective analysis are still valid in 2019. HFrEF treatment changed because of results of EMPHASIS-HF published in 2011 and the PARADIGM-HF published in 2015. Why did the authors choose to analyze the effect of beta blocker treatment on 365 days all-cause mortality in real-world patients 2007-2013?

Why did the authors decide to measure the first hospitalization independent whether it was a hospitalization for heart failure or other reason? Reason for this question is the fact that it is clear that beta-blocker treatment should reduce heart failure related-hospitalization but not hospitalization for other comorbidity. How can You exclude that the effect of beta-blocker treatment on heart failure related hospitalization was underestimated?

Why did the authors did not adjust for other heart failure drug treatment?

6. PLOS authors have the option to publish the peer review history of their article (what does this mean?). If published, this will include your full peer review and any attached files.

Reviewer #1: No

Reviewer #2: Yes: Roger Hullin MD

---

## [Author Response · Author response to Decision Letter 0]

25 Mar 2020

We have responded to the reviewer comments in the "Response to reviewers" file.

---

## [Decision Letter · Decision Letter 1]

22 Apr 2020

PONE-D-19-24254R1

Evidence-based beta blocker use associated with lower heart failure readmission and mortality, but not all-cause readmission, among Medicare beneficiaries hospitalized for heart failure with reduced ejection fraction

PLOS ONE

Dear Dr. Loop,

Thank you for submitting your manuscript to PLOS ONE. After careful consideration, we feel that it has merit but does not fully meet PLOS ONE’s publication criteria as it currently stands. Therefore, we invite you to submit a revised version of the manuscript that addresses the points raised during the review process.

One of the reviewers was still not satisfied with the paper. This is why we internally discussed the issue. The major concern of the reviewer remains the interpretation of the early effects on mortality (within 30 days). I understand that the data do not allow to investigate the exact reason for this effect and that the interpretation remains somewhat speculative. Nevertheless, I agree with the reviewer that overall better compliance may play an important role and would like to ask you to further emphasize this possibility in your limitation and include this also in your conclusions.

On purpose, the reviewer's comments are not included in this mail as they do not provide additional information.

We would appreciate receiving your revised manuscript by Jun 06 2020 11:59PM. To enhance the reproducibility of your results, we recommend that if applicable you deposit your laboratory protocols in protocols.io, where a protocol can be assigned its own identifier (DOI) such that it can be cited independently in the future. For instructions see: http://journals.plos.org/plosone/s/submission-guidelines#loc-laboratory-protocols

We look forward to receiving your revised manuscript.

Kind regards,

Hans-Peter Brunner-La Rocca, M.D.

Academic Editor

PLOS ONE

---

## [Author Response · Author response to Decision Letter 1]

24 Apr 2020

Thank you very much for the opportunity to revise and resubmit our manuscript. Upon further considering of the comments from one reviewer and yourself, we whole-heartedly agree that our inability to separate the benefits of beta blockers from general adherence, especially to other therapies, is a limitation. Therefore, we have added a statement regarding this limitation in the Discussion, as well as in our Conclusions. We have reproduced both sets of text below for your convenience.

Discussion

…

Another source of confounding could have been that patients who are more likely to be adherent to any drug regimen for HFrEF and thus have better outcomes may be more likely to fill a prescription for a beta blocker. Therefore, this analysis may overestimate the apparent benefits of beta blocker use. In particular, filling a prescription for a beta blocker shortly after discharge may be a marker of adherence to medical therapy in general, which may help explain the large apparent benefit on mortality immediately following hospitalization.

Conclusions

However, in this retrospective observational study, we were unable to disentangle the effects of beta blocker use from other characteristics and behaviors such as adherence to other therapies. This adherence to other therapies may have led to overestimation of the protective associations of beta blocker use, particularly on mortality.

We very much hope these changes are satisfactory, as we would be proud to have this manuscript published in PLOS ONE.

---

## [Editor Report · Decision Letter 2]

30 Apr 2020

Evidence-based beta blocker use associated with lower heart failure readmission and mortality, but not all-cause readmission, among Medicare beneficiaries hospitalized for heart failure with reduced ejection fraction

PONE-D-19-24254R2

Dear Dr. Loop,

We are pleased to inform you that your manuscript has been judged scientifically suitable for publication and will be formally accepted for publication once it complies with all outstanding technical requirements.

With kind regards,

Hans-Peter Brunner-La Rocca, M.D.

Academic Editor

PLOS ONE

---

## [Editor Report · Acceptance letter]

26 Jun 2020

PONE-D-19-24254R2 

Evidence-based beta blocker use associated with lower heart failure readmission and mortality, but not all-cause readmission, among Medicare beneficiaries hospitalized for heart failure with reduced ejection fraction 

Dear Dr. Loop:

I'm pleased to inform you that your manuscript has been deemed suitable for publication in PLOS ONE. Congratulations! Your manuscript is now with our production department. 

Kind regards, 

on behalf of

Dr. Hans-Peter Brunner-La Rocca 

Academic Editor

PLOS ONE